# Assessment of end-of-life vehicle recycling: Remanufacturing waste sheet steel into mesh sheet

Ziyad Tariq Abdullah ●*

Mechanical Techniques School, Institute of Technology Baghdad, Middle Technical University, Baghdad, Iraq

* ziyad_tariq12@mtu.edu.iq

**Data Availability Statement:** All relevant data are within the manuscript.

**Funding:** The author received no specific funding for this work.

## Abstract

The automobile industry contributes significantly to global energy use and carbon emissions. Hence, there are significant economic and environmental benefits in recovering materials from end-of-life vehicles (ELVs). Here, the remanufacturing of waste steel sheet (WSS) from ELVs into useful mesh steel sheet (MSS) for metal forming applications was evaluated based on its technological, economic, and environmental feasibility. A remanufacturing plant with a dismantling capacity of over 30,171 ELV/year and a recovery capacity of 1000 $m^2$/d of WSS was used as a case study. Remanufacturing can achieve a total reduction of ~3800 kg $CO_2$/ELV and an economic benefit of ~775 USD/ELV compared with conventional recycling. The calculated feasibility indexes were similar to or exceeded standard feasibility thresholds, indicating that WSS remanufacturing is a viable sustainable development route and has synergistic benefits when combined with existing recycling plants, especially in developing countries as small-to-medium enterprises.

## Introduction

The global automotive industry currently has a large negative environmental impact due to high $CO_2$ emissions, high energy use, and a large waste stream of end-of-life vehicles (ELVs). The extensive use of steel and other metals in automobile bodies results in a waste stream of ELVs with a high embodied energy (25–28 MJ/kg) [1]. Currently, 1200 kt/year of waste steel sheet (WSS) from automotive bodies is generated in the United States alone, which is expected to increase to ~125 kt/year by 2035 and 246 kt/year by 2050 [2].

To recover some of the embodied energy in automobile parts, there is a large global industry based on recycling ELVs, which reduces the overall energy consumption and $CO_2$ emissions of the automotive industry via effective material flow. The effective management of ELV recycling can include: the reuse of functional parts to reduce the energy and $CO_2$ emissions related to producing new spare parts; processing recyclable materials to produce raw materials for other processes; and energy recovery and thermal energy generation from automotive shredder residue (ASR), which contains non-recyclable materials such as glass fibre, polymers, and glass [3]. The energy use and $CO_2$ emissions related to the production of new vehicles can

**Competing interests:** The authors have declared that no competing interests exist.

be reduced when the aluminium, steel, and/or plastics in the ASR are recycled. However, material recycling rates are still very low, e.g., ~17% [4], which is far from achieving a circular economy (CE) in the automotive industry. Effective CE strategies that enable a higher rate of scrap utilization require a more efficient use of ELV materials and new car designs that facilitate reuse and remanufacturing. Emerging vehicle technologies, such as electric cars and lightweight vehicles, should follow CE design criteria to increase scrap utilization in the future. This is especially important considering the increasing use of composite materials in modern automobiles, which have less mature recycling technologies [4]. An economic feasibility analysis indicated that dismantling of ELVs in Korea accounted for the largest amount of material in the recycling process [5]. In general, new technologies are required to reduce the recycling cost, which can prevent material being sent to landfill for economic reasons. In addition, policies should be developed to provide financial support to ELV dismantlers to increase the overall recycling rate. At the post-dismantling stage, greater attention should be paid to the shredding step and diversifying the treatment methods for recycled ASR. Institutional or financial support will be essential to assist with the initial investment costs for developing ASR treatment technologies and constructing new facilities to increase the rate of scrap utilization. In the case of steel, optimization of ASR recycling to minimize the amount of waste disposed in landfill can result in an energy reduction of 21,100 MJ/vehicle and reduction in carbon emissions of 271 kg $CO_2$/vehicle [6]. Hence, it is clear that there are significant energy savings and environmental benefits that could be achieved by optimising the ELV recycling process.

Currently, WSS from ELVs is not remanufactured on an industrial scale. Most sheets are subjected to shredding and smelting processes to recover the steel, or are disposed of in landfill. As an alternative to reuse and shredding, remanufacturing can be used to upcycle the recovered materials, such as the WSS, into value-added products. Previous studies showed that remanufacturing is economically viable for the sustainable management of ELV recycling factories [7, 8]. The application of remanufacturing to process the WSS into other useful products could lead to an energy reduction of ~25 GJ/vehicle [6] as it has a much lower energy consumption than recycling, which requires that the WSS is shredded and smelted, and needs metallurgical infrastructure. This corresponds to a reduction of ~1119 kg $CO_2$/vehicle, in addition to the environmental benefits of diverting materials from landfill. Furthermore, remanufacturing recovers the embodied energy and produces a profitable product to enhance the economic viability of the process.

The current study proposes a new remanufacturing process for integration with ELV recycling, which involves remanufacturing WSS from the exterior components into mesh sheet steel (MSS). This remanufacturing process is expected to be an environmentally conscious strategy to reduce the amount of useful material being shredded and sent to landfill, while greatly reducing the energy consumption of the process compared to recycling. Furthermore, the MSS product can be used for a wide range of sheet metal forming processes and has a market value that could provide an additional income to the recycling plant. Here, the WSS–MSS remanufacturing process is evaluated as a triple-bottom-line solution for ELV recycling plants, where the technical, economic, and environmental feasibility is quantified based on data from the literature.

## Methods

### Remanufacturing process

In this study, technical feasibility of the remanufacturing process is proved based on a laboratory scale and the results are used to develop an analysis model. Remanufacturing units were designed to be matched with disassembly lines of end-of-life vehicles, which could be applied

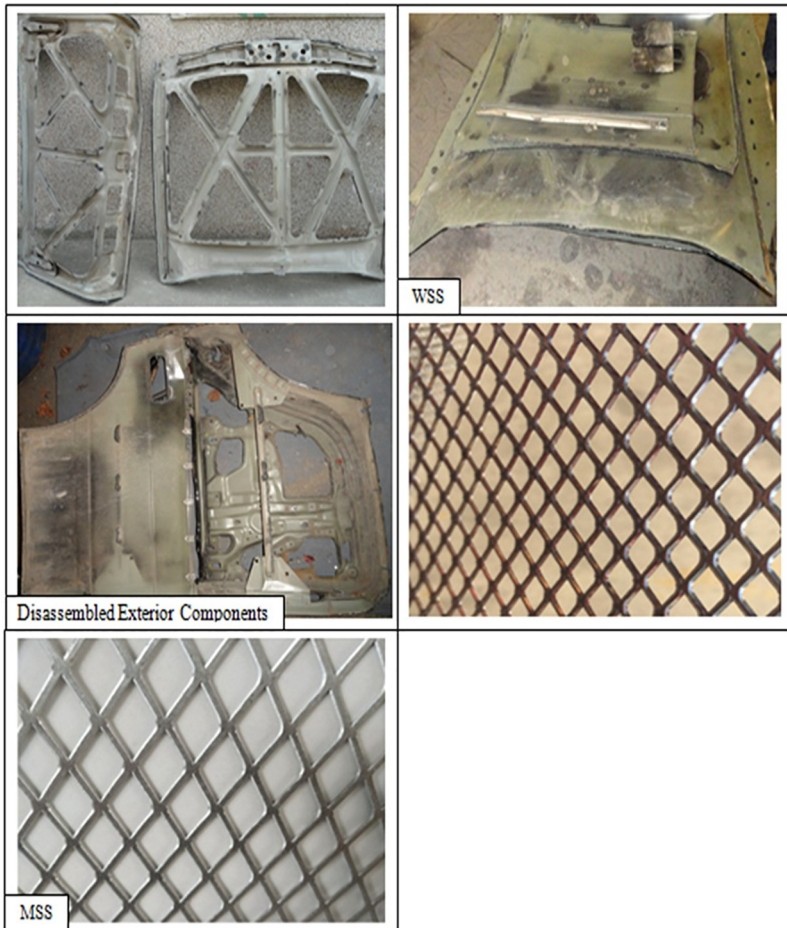

**Fig 1. WSS–MSS remanufacturing method.** Schematic diagrams showing the production of mesh from flat steel sheets.

worldwide [9, 10]. In addition, scientific literature based on case studies from Europe, China, Malaysia, Japan, Turkey, Australia, Korea, Belgium, Taiwan, and the Netherlands was considered to develop the feasibility indicators. To ensure that the remanufacturing technique is sufficiently sustainable, the unit cell of machines should be matched with existing dismantling plants. The unit cell of machines includes: a hydraulic alligator scrap-metal cutting machine, a CNC plasma cutting machine, an expanding machine, a flattening machine, and a press-break machine. A unit cell can represent a small-to-medium-sized enterprise with the ability to be up-scaled into a large company, with the aim of processing the total amount of generated end-of-life cars worldwide using many such plants.

The remanufacturing process for converting WSS from ELVs into MSS is shown in Fig 1. The first step in the process is pre-disassembly, which is performed either manually or using machines on a disassembly line along which ELVs move and stop at different stations. The vehicle is disassembled using a hydraulic alligator scrap-metal cutting machine, or similar. Post-disassembly is the process of cutting the WSS from the frames of the exterior components using CNC plasma, flame, laser, or water-jet cutting machines (hereafter referred to as the cutting process/machines). This is followed by expanding, where the expanding machine cuts slots into the WSS, and then stretch the sheet perpendicular to the slot length to open the slots

and increase the surface area of the WSS to produce the MSS. Then, the expanded MSS is flattened by a roller, which can further increase its surface area. Finally, levelling is used to straighten the MSS to make it suitable for sheet metal forming applications, such as tubes, fences, sandwich panels, air cooler frames, corrugated panels, electrical connectors, and a wide range of decorative applications. The MMS can be cut to size for a specific application by the shear-braking method.

The machines for CNC plasma or laser cutting, expanding, flattening, levelling, and shear braking are non-energy-intensive technologies compared with the recycling process of smelting steel. In addition, they are well-established and common technology for integration into existing disassembly plants. Considering the proposal of remanufacturing plants as small-to-medium enterprises in developing countries, such technology is much more appropriate than high-temperature smelting furnaces for recycling, considering the capital cost and energy usage.

During the expansion process, there is an increase in the surface area of the WSS, which depends on the mesh expansion and flattening parameters. The change in the length of the metal sheet after expanding into mesh is described by Eq (1), where $L_e$ is the final expanded length (m), $L_s$ is the initial sheet length (m), $T_s$ is the sheet/mesh thickness (mm), and SWP is the short way pitch and SWDT is the strand width, as defined in Fig 2.

$$L_e = [0.5L_s/T_s] * [(\text{SWP-SWDT})] \tag{1}$$

For example, for an initial sheet with a thickness of 1 mm and length of 5 m, and a mesh hole with a long way pitch (LWP) = 5 mm, SWP = 3.7 mm, and SWDT = 0.5 mm, the length after expansion is 8 m. The expansion ratio (*ER*) is defined by Eq (2), while the area gain ratio (*AR*) is defined by Eq (3), where $A_{WSS}$ and $A_{MSS}$ are the areas of the WSS and MSS,

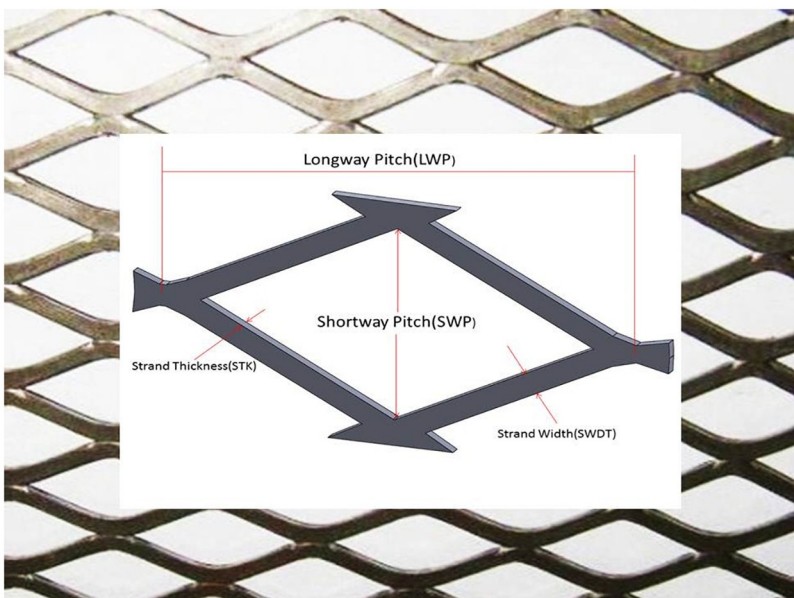

**Fig 2. Schematic of the geometry of a single MSS hole.**

respectively.

$$ER = L_e/L_s \qquad (2)$$

$$AR = A_{WSS}/A_{MSS} \qquad (3)$$

In the case study considered here, a constant-cycle-time disassembly line was assumed, with an operation period of 250 d/year and zero-queue operation with four parallel pre-treatment stations (20 min processing time each). The highest productivity of this disassembly line is 30,171 ELV/yr (or 121 ELV/d) with a mean cycle time of 7.16 min/ELV [9, 10]. This corresponds to 1,882 t/year of WSS, or 1000 m$^2$/d of WSS to be recovered and remanufactured into 5,987–11,592 m$^2$ of MSS. Hence, 500 doors/d will need to be pre-processed. A hydraulic alligator scrap-metal cutting machine (1200-mm blade length, 750-mm blade opening, and 7.5-cycle/min shearing frequency) has a capacity of 7200 door/day, so one machine is sufficient for the proposed plant. The cutting machines can disassemble 168 doors/d. For example, a CNC plasma cutting machine with five cutting heads and a work area of $24 \times 7$ m$^2$ can recover 1 m$^2$ of WSS in 1 min. The remanufacturing of WSS to MSS needs to be optimized so that each 1 m$^2$ of WSS (7.8 kg) only requires 1 min to be recovered, and 2–3.5 min to be expanded into MSS for later flattening by the rolling machine.

The economic feasibility can be defined considering the ability of remanufacturing to increase the recovery rate compared with recycling, since both have the same percentage of metal recovery (~82%) [11]. Recycling of 1 kg steel can give an income of 0.2–0.25 USD/kg, while remanufacturing can give an income of 3.21 USD/kg in the form of MSS. The objective of adding remanufacturing to the dismantling plant is to minimize the recycling cost by optimizing the number and location of vehicle dismantling stations. Here, 250 ELV/d being dismantled at 11 different stations is considered optimal.

## Calculation of the feasibility indexes

Sustainability assessments involve triple-bottom-line analyses that include feasibility indexes related to the technical, economic, and environmental factors [12–16]. Experience-based, literature-based, and scenario-based analyses were applied to select the relevant parameters and weights for the current study. The calculations of the indexes were based on equations given in previous studies [7–11, 17–24], as described in the following sections.

**Technical feasibility index.** To analyse the technical feasibility of the proposed remanufacturing method, the technical feasibility index ($T$) was calculated. Here, $T$ is defined as the mean of the feasibility indexes for pre-disassembly, post-disassembly, expanding, flattening, levelling, and shear-braking processes. Each one of these indexes has a weight of importance ($W$) corresponding to the average of the constituent factors.

The pre-disassembly feasibility weight ($W_{EPrD}$), post-disassembly feasibility weight ($W_{EPoD}$), and expanding feasibility weight ($W_{EEx}$) were calculated as the weighted averages for the exterior components listed in Table 1, multiplied by the overall pre-disassembly weight ($W_{PrD} = 0.85$), overall post-assembly weight ($W_{PoD} = 0.95$), and overall expanding weight ($W_{Ex} = 0.95$), respectively, as shown in Eqs (4), (5) and (6), respectively. The weight values were determined using a fuzzy logic method by considering the ease with which the components could be disassembled and processed. Taking data from the literature [13–16], a scale with values of [0.4, 0.45, 0.5, 0.6, 0.65, 0.75, 0.80, 0.85, 0.95] was used, where higher values indicate a higher level of importance and applicability. One of these values was assigned to both the importance and applicability for each remanufacturing parameter or exterior component,

**Table 1. Calculated weights of the individual components used to calculate $W_{EPrD}$, $W_{EPoD}$, and $W_{EEx}$.**

| Component | Abbreviation | Pre-disassembly weight | Post-disassembly weight | Expanding weight |
|---|---|---|---|---|
| Front Door | $W_{FD}$ | 0.75 | 0.65 | 0.85 |
| Rear Door | $W_{RD}$ | 0.70 | 0.55 | 0.85 |
| Hood | $W_{H}$ | 0.90 | 0.75 | 0.85 |
| Roof | $W_{R}$ | 0.95 | 0.95 | 0.95 |
| Boot | $W_{B}$ | 0.85 | 0.35 | 0.85 |
| Front Fender | $W_{FF}$ | 0.45 | 0.40 | 0.75 |
| Rear Fender | $W_{RF}$ | 0.40 | 0.35 | 0.85 |

and these values were multiplied to obtain the weight.

$$W_{EPrD} = [W_{PrD}[W_{FD} + W_{RD} + W_{H} + W_{R} + W_{B} + W_{FF} + W_{RF}]]/7 = 0.607 \qquad (4)$$

$$W_{EPoD} = [W_{PoD}[W_{FD} + W_{RD} + W_{H} + W_{R} + W_{B} + W_{FF} + W_{RF}]]/7 = 0.559 \qquad (5)$$

$$W_{EEx} = [W_{Ex}[W_{FD} + W_{RD} + W_{H} + W_{R} + W_{B} + W_{FF} + W_{RF}]]/7 = 0.794 \qquad (6)$$

Similar to the other indexes, the flattening weight ($W_{EF}$) was calculated by multiplying the overall flattening weight ($W_{FI} = 0.85$) by the flattening performance of the component ($W_{FP} = 0.95$), as shown in Eq (7). Similarly, the levelling weight ($W_{EL}$) was calculated by multiplying the overall levelling weight ($W_{LI} = 0.85$) by the levelling weight of the component ($W_{LP} = 0.95$), as shown in Eq (8). Finally, the shear-braking weight ($W_{ESB}$) was calculated by multiplying the overall shear-braking weight ($W_{SBI} = 0.5$) by the shear-braking weight of the component ($W_{SBP} = 0.95$), as shown in Eq (9). Here, the same exterior components listed in Table 1 were considered.

$$W_{EF} = W_{FI} \times W_{FP} = 0.8075 \qquad (7)$$

$$W_{EL} = W_{LI} \times W_{LP} = 0.8075 \qquad (8)$$

$$W_{ESB} = W_{SBI} \times W_{SBP} = 0.475 \qquad (9)$$

Finally, *T* was calculated by taking the average of the indexes for each process, as shown in Eq (10), giving a final value of 0.675.

$$T = (W_{EPrD} + W_{EPoD} + W_{EEx} + W_{EF} + W_{EL} + W_{ESB})/6 \qquad (10)$$

**Economic feasibility index.** The economic feasibility analysis assumed that 30,171 passengers cars can be dismantled for recycling every 250 d [2, 6, 25], corresponding to 121 ELV/d being remanufactured. In this calculation, it was assumed that a typical ELV has an average weight of ~1 t/ELV, where net weight of steel is 754 kg/ELV [4, 6]. If the maximum dismantling and sorting capacity of the plant is 4000 t/week, then 4215 vehicles can be processed per week. It should be noted that of the 754 kg of steel, only 62.4 kg of this is WSS (considered during remanufacturing calculations), while the remainder of the steel can be recycled (considered in the recycling case study). Therefore, 6 plasma cutting machines and 14 expanding machines are required to process the 6744 m$^2$ of WSS to produce 40059–77556 m$^2$ of expanded MSS.

The direct sale of expanded MSS as a raw material has a unit price of 5 USD/m² giving a direct lower bound (DLB) of 28,750 USD/d and a direct upper bound (DUB) of 55,660 USD/d. The indirect sale of expanded MSS as a finished product has a unit price of 10 USD/m², resulting in an indirect lower bound (ILB) and indirect upper bound (IUB) of 57,500 and 111,320 USD/day respectively. The economic index (C) was calculated based on the direct and indirect sale of MSS, as shown in Eq (11), where $N_s$ is the number of 8 h shifts (3 in this case) and $C_T$ and is the total cost (20,427 USD), giving a value of 0.834.

$$C = [(N_s \times DLB) - (C_T)/(DLB)] + [(N_s \times DUB) - (C_T)/(DUB)] + [(N_s \times IDLB) - (C_T)/(IDLB)] + [(N_s \times IDUB) - (C_T)/(IDUB)]/4 \tag{11}$$

**Environmental feasibility index.** The exact quantity of $CO_2$ emissions avoided by a specific remanufacturing process depends on the comparisons of WSS–MSS remanufacturing with manufacturing of virgin steel sheet (VSS) into MSS. The environmental feasibility index (E) was based on a remanufacturing unit with a productivity of 5,750–11,132 m²/d of MSS, as follows. Here, the eco-cost saving ($\varepsilon_{ec}$; USD/d) is defined, which is the carbon offset costs avoided by the reduced energy consumption and carbon emissions by the remanufacturing process, as given by Eq (12), where $M_{VSS}$ is the weight of VSS avoided by recycling or remanufacturing (kg), and $\varepsilon_{VSS}$ is the corresponding eco-cost saving. A similar equation is given in Eq (13) for the $CO_2$ saving ($\varepsilon_{CO2}$; $kg_{CO2}$), where $M_{CO2}$ is the amount of $CO_2$ produced per 1 kg of VSS. The environmental feasibility index (E) was calculated using Eq (14), where the subscripts 'rem' and 'rec' refer to the remanufacturing and recycling cases, respectively, and these ε values were calculated using Eq (12) for the respective processes (where $\varepsilon_{rem}$ or $\varepsilon_{rec}$ were substituted for $\varepsilon_{ec}$ in this calculation). The $M_{VSS}$ value for recycling was 62.4 kg (assuming that the WSS is used for mesh-steel applications). The $M_{VSS}$ for remanufacturing has a range of 62.4–698 kg. This corresponds to the range of the upper and lower bounds, where the upper bound assumes that the steel is used for sheet-metal products and that the eco-design standards are satisfied. The $\varepsilon_c$ of VSS is 0.55 USD/kg and its $M_{CO2}$ is 1.559 $kgco_2/kg_{VSS}$. These values were the same for both remanufacturing and recycling, as they correspond to VSS and are independent of recycling/remanufacturing conditions. The E index was calculated as the average of the ε values determined for the upper and lower bounds of productivity for the remanufacturing process.

$$\varepsilon_{ec} = M_{VSS}\varepsilon_{VSS} \tag{12}$$

$$\varepsilon_{CO2} = M_{VSS}M_{CO2} \tag{13}$$

$$E = (\varepsilon_{rem} - \varepsilon_{rec})/\varepsilon_{rem} \tag{14}$$

**Sustainability index.** The viability of the remanufacturing process was evaluated based on a sustainability index (SI), calculated using the following Eq (15). SI is always between 0 and 1, where a higher value indicates a more sustainable process.

$$SI = W_T T + W_C C + W_E E \tag{15}$$

Here, $W_T$, $W_C$, and $W_E$ are the weights of importance of T, C, and E, respectively, in determining the sustainable development decision.

## Results

### Global potential for remanufacturing

This section discusses the global potential for remanufacturing of WSS from ELVs. The material savings due to the proposed remanufacturing process are shown in Table 2. The numbers of ELVs disposed of each year in various countries were taken from the literature. The available WSS was calculated assuming that an average of 80 m$^2$ can be recovered from each car to prevent manufacturing of ~698 kg of new VSS The added value by remanufacturing the sheet was calculated assuming that 1 m$^2$ of WSS can be remanufactured into 5.94–11.5 m$^2$ of MSS. For example, 6.3 and 9 million vehicle/year were processed in Europe in 2009 and 2012, respectively [17]. This corresponds to ~50–72 million m$^2$ of WSS in the form of exterior components, such as the bonnet, hood, roof, front and rear doors, and fenders. Assuming the upper bound of MSS production, 103 million m$^2$ of MSS could be remanufactured, preventing the production of 8 million t of new steel, corresponding to cost savings of 454 million USD and reduced emissions of 1.2 million t of $CO_2$. Hence, there is clearly a large market and potential for steel remanufacturing.

Increasing the weight percentage of steel in the recycling waste stream from 62% to 73% [17] is considered an environmentally conscious approach, where the exterior components account for ~6% (62.4 kg) of the total steel weight from the ELV. Introducing remanufacturing to the pre-shredder treatment can recover at least 6–8 m$^2$ of WSS from the exterior components (avoiding the production of the same amount of VSS), which can be converted into ~20–90 m$^2$ of MSS.

In Europe, 1.93–2.34 Mt/year of ASR waste is produced, accounting for up to 10% of the total hazardous waste and ~60% of the total shredder waste [22]. This large volume, which is 20–25% of the average ELV weight, is due to the complexity of ASR recycling. Hence, remanufacturing can provide a possible solution for optimizing ELV recycling systems. Remanufacturing portfolios should be integrated with recycling and landfill disposal strategies by conducting research to innovate new methods of reuse, recycling, and remanufacturing of ELV waste. Remanufacturing of WSS can change the processing model currently used in the US, which has recycling rates of ~75–89% [22]. While all of the material is shredded, there is ~10% loss of recyclable materials during processing, leading to a maximum recycling rate of ~90%. The income provided by WSS remanufacturing could be exploited to promote the recycling of tires and other non-metal materials, which would greatly contribute to meeting the

**Table 2. Calculated material savings due to the proposed remanufacturing process.**

| Region | Number of ELVs (million/year) | Available WSS (million m$^2$/year) | Added value by remanufacturing (million m$^2$/year) |
|---|---|---|---|
| Global [26] | 100 | 800 | 4752–9200 |
| Europe [27] | 17 | 136 | 808–1564 |
| China [28] | 14 | 112 | 665–1288 |
| Malaysia [29] | 6.7 | 53 | 317–613 |
| Japan [23] | 5 | 40 | 238–460 |
| Turkey [30] | 3 | 24 | 143–276 |
| Australia [31] | 0.6 | 4.9 | 29–56 |
| Korea [23] | 0.5 | 4.2 | 25–48 |
| Belgium [31] | 0.4 | 3.2 | 190–368 |
| Taiwan [23] | 0.27 | 2.2 | 13–25 |
| Netherlands [23] | 0.23 | 1.9 | 11–22 |

European goal an ELV recycling efficiency of 85% (without energy recovery), while the goal of 95% recovery of the average ELV weight can only be achieved with improved separation and sorting technology [22].

## Comparison of reuse, recycling, and remanufacturing

Reuse, recycling, and remanufacturing are defined with respect to the aims of the current study. In the case of reuse, a limited 6–8 $m^2$/car of WSS is available to be used directly for metal-forming purposes to manufacture new products to prevent the production of the same amount of new sheet steel. In the case of recycling, the 6–8 $m^2$/car of WSS is converted into ASR to be prepared for smelting. Finally, in the case of remanufacturing, the 6–8 $m^2$/car of WSS is processed into MSS to be used directly for metal-forming purposes to manufacture new products and prevent production of an average of 80 $m^2$ of new MSS (where the increase in area is achieved via the expanding process).

Based on the average weight (~755 kg) of a hulk vehicle for recycling, ~21% can be stripped, ~42% can be reused, ~51% can be recycled, ~8% can be recovered, and ~1.4–6% is sent to landfill [6]. During the recycling process, the weight percentage of steel in the recycling stream is continuously increased as the metals are separated from the waste materials: (1) dismantling of the ELVs; (2) sorting of recyclable materials; (3) processing of recyclable materials. In conventional ELV recycling, the steel percentage is ~50% in phase (1), and ~60% in (2) and (3) [6]. In contrast, for ELV recycling with remanufacturing, the steel content in (3) can be increased to close to 100%. Steel is 100% recyclable, so remanufacturing combined with reuse satisfies the legislation of the European Union, which states that a minimum of 85% of ELV weight has to be reusable and recyclable materials, combined with energy recovery from another 10%, giving a total recycling efficiency of 95% [6]. Conventionally, the materials contained in an ELV are classified into reusable components, recyclable materials, and ASR. By applying WSS remanufacturing, the percentage of useful steel content can be increased compared to recycling alone. The produced MSS can be used for various metal forming purposes, such as producing tubes, fences, boxes, and decorative items, eliminating the need to use VSS. Table 3 compares the efficiency ($\eta$) of dismantling the ELVs and selling the exterior components for reuse with the remanufacturing of out-of-date parts (that cannot be reused) into value-added products. These efficiencies were calculated using values from the literature [6] and by comparing the potential for remanufacturing and reuse to increase the weight percentage of steel in the recycling waste stream.

Remanufacturing 1 $m^2$ (7.8 kg) of WSS will prevent the production of the same weight of new steel. Table 4 compares the different recovery techniques based on the weight of steel recovered, and the potential energy and $CO_2$ reductions [6]. Clearly, landfill does not allow the recovery of any steel. Although recycling can recover the most steel, the WSS–MSS

**Table 3. Comparison of the calculated efficiency of reuse and remanufacturing of exterior components.** Adapted from [6].

| Exterior Component | $\eta$ (reuse) | $\eta$ (reman.) |
|---|---|---|
| Front Fender | 1 | 1.115 |
| Front Door | 0.64 | 0.75 |
| Rear Door | 0.62 | 0.74 |
| Boot | 0.85 | 0.96 |
| Hood | 0.80 | 0.92 |
| Back Fender | 0.74 | 0.85 |
| Roof | 0.88 | 1 |

**Table 4. Calculated weight of recovered steel using different recovery techniques.** Adapted from [6].

| Recovery technique | Weight of recovered steel (kg) | Energy consumption reduction (MJ/vehicle) | $CO_2$ reduction (kg$_{CO2}$/vehicle) |
|---|---|---|---|
| Landfill | 0 | N/A | 1747 |
| Sorting for energy recovery | 253 | N/A | 4368 |
| Recycling | 818.6 | 7535 | 28200 |
| WSS disassembly | 62.4 | 1747 | 7535 |
| WSS–MSS remanufacturing | 156–718 | 4368–28200 | 25116 |
| Component reuse | 340 | 25116 | 4368 |

remanufacturing process can recover nearly as much as recycling (at the upper processing limit), but with added energy savings and economic benefits compared to recycling. This can prevent the emission of 1.559 kg of $CO_2$ and result in a cost saving of 0.476 USD/kg [23]. As ELVs primarily contain steel, WSS remanufacturing could result in energy savings of 2.7–12.558 GJ/ELV in addition to the energy reductions of 39.9, 34.4, and 3.9 GJ attributed to material recycling, component reuse, and energy recovery techniques, respectively [6].

The unit prices of various exterior components are shown in Table 5 for reuse and remanufacturing. The remanufacturing values were calculated by multiplying the area by the AR (lowest and highest values of 4.5 and 11.5, respectively) and by the price per area (lowest and highest values of 4.5 and 10 USD/m$^2$, respectively). The sale of a component for reuse is only possible when the piece is new enough to be compatible with cars in service, while recycling and remanufacturing are the only options at the end-of-life phase of a specific model of car. When the exterior components cannot be sold as spare parts, they can be remanufactured into MSS. It can be seen that, for nearly all of the components, a higher sales price can be achieved by remanufacturing, even at the lowest AR, while the highest AR ensures much higher returns. The remanufacturing/reuse (R/R) ratios of the components are also shown in the table, which were calculated by dividing the MSS price (lower bound) by the component reuse price. It can be seen that R/R varies (~0.864–1.7). The hood has a low R/R ratio as a similar sales price is obtained for reuse or remanufacturing at the lowest assumed AR. The roof sheet is usually considered as a waste piece and can account for more than 20% of the total WSS weight. As it has a zero reuse value, and R/R could not be calculated. However, the roof has high market price (322 USD) due to its high area, and its remanufacturing is particularly economically feasible. The cost of an ELV is ~115–175 USD/t, while scrap steel can be sold in the market for ~437 USD/t [9]. Hence, remanufacturing WSS into MSS can earn 100–460 USD/ELV more than the sale of dismantling plants.

The profit of vehicle recycling over the planning horizon can be maximized by optimizing the: income from the isolated metals; cost for thermal treatment of ASR; ASR landfill disposal cost; procurement cost of the vehicle; vehicle storage cost; processing, fragmentation, and sorting costs; transportation costs for the components, remanufactured steels and shredded

**Table 5. Calculated unit price and R/R ratio of exterior components, reusing compared remanufacturing.** Adapted from [9].

| Exterior component | Area (m$^2$) | Sales price, reuse (USD) | Sales price, remanufacturing lower bound (USD) | Sales price, remanufacturing upper bound (USD) | R/R |
|---|---|---|---|---|---|
| Hood | 1 | 28.94 | 25 | 115 | 0.864 |
| Front Door | 1 | 43.41 | 25 | 115 | 0.576 |
| Boot | 1 | 14.47 | 25 | 115 | 1.728 |
| Rear Door | 1 | 36.18 | 25 | 115 | 0.691 |
| Roof | 2.8 | 0 | 70 | 322 | N/A |

material. The cost of ASR landfill disposal depends on various factors [29], including the landfill costs and local taxes. The landfill costs depend on the specific density of the waste. For a density of 300–350 kg/m$^3$, the cost can be 36.2–53.8 USD/t. The non-metal fraction of the ASR has the lowest disposal cost (42 USD/t), followed by the second non-metal fraction (32 USD/t), non-ferrous mix (42 USD/t), rubber-plastic-rubber fraction (33 USD/t), and insulated copper wires (209 USD/t). There has been a continuous increase in the cost of ASR landfill disposal, e.g., increases from 30.5, to 43.2–50.4, and 60.4 USD/t) in 2012, 2013, and 2014, respectively, in Europe [22]. Further, the same study showed that local taxes on polluting activities can be as high as 52 USD/t.

The shredding, separation, and sorting efficiency are generally ~87%, while the corresponding energy recovery efficiency can reach ~11%, depending on the technology used. The use of remanufacturing reduces the efficiency of these processes to ~66%, but results in a higher total efficiency of recycling (less material sent to landfill). Assuming a WSS mass of 62.4 kg from the exterior components of an ELV, the power consumption and productivity of recycling with and without remanufacturing were calculated. The power consumption of (i) shredding, separation, and sorting of 1 ELV is 220 kWh, while for remanufacturing with the (ii) lowest and (iii) highest ER values the power consumption is 180 and 230 kWh, respectively. The productivity for cases (i), (ii), (iii) are 11, 11.25, and 128 t/h, respectively. In the case of recycling with remanufacturing, the embodied energy reduction is equal to 62.4 kg × 20.1 MJ/kg = 1254.24 MJ [1, 23]. However, in the case of recycling without remanufacturing this value is zero as this is the reference case. Table 6 compares the direct and indirect reductions in material, energy, $CO_2$ emissions, and cost (per vehicle) for recovering WSS from ASR (to prepare it for recycling by smelting) or remanufacturing into MSS. It can be seen that remanufacturing greatly enhances the reductions of all environmental metrics.

The amount of material sent to landfill is ultimately an economic decision, which is defined by the following threshold: $ATT_{Cost} \geq 94.8$ USD/t or $MSWI_{Cost} \geq 81.8$ USD/t [22], where ATT is advanced thermal treatment and MSWI is a municipal solid waste incinerator. Two policies can be applied to reduce the amount of ASR sent to landfill: (i) increasing the price of landfill disposal; and/or (ii) use of remanufacturing so that landfill disposal threshold becomes higher (e.g., $ATT_{Cost} \geq 119.8$ USD/t or $MSWI_{Cost} \geq 106.8$ USD/t). Introducing policies to increase landfill costs is expected to have limited effect, so the introduction of remanufacturing of ELVs is proposed as a more effective triple-bottom-line approach. To increase the recovery of embodied energy and reduce the cost of landfill disposal paid by the recycling plant, the amount of ASR send to landfill should be less than 5–15% of the total vehicle weight. Larger amounts of ASR are sent to landfill when MSWI and ATT systems are used, resulting in higher costs. The recycling quota can be increased to ~3% by the introduction of remanufacturing to provide the required economic resources for material and energy recovery.

Table 7 compares the operational costs (USD/t scrap metal) for the various steps of producing ASR for recycling and landfill [22], and for those of the proposed remanufacturing process. The total operational cost of the remanufacturing process is 3–10 times lower than that of

**Table 6. Comparison of the calculated direct and indirect reductions in material, energy, $CO_2$ emissions, and cost for recycling or remanufacturing WSS (per vehicle).**

| Environmental Indicator | Direct effect of remanufacturing | Indirect effect of remanufacturing | Direct effect of recycling |
|---|---|---|---|
| Reduction in material (kg) | 371–718 | 62.4 | 62.4 |
| Reduction in power (kWh) | 1,112–2,153 | 182 | 0.78 |
| Reduction in $CO_2$ emissions ($kg_{CO2}$) | 477–924 | 78.1 | 0.3 |
| Cost savings (USD) | 697.5 | 349 | 74.8 |

**Table 7. Comparison of the calculated costs of ASR separation and sorting with that of WSS remanufacturing.** Adapted from [22].

| ASR | Cost (USD/t) | WSS remanufacturing | Cost (USD/t) |
|---|---|---|---|
| Shredding | 26.3 | Hydraulic alligator cutting | 1.8–8.2 |
| Magnetic separation | 9.4 | Plasma cutting | 2.9–13.3 |
| Horizontal separation | 33.4–74.6 | Sheet expansion | 1.9–8.7 |
| Manual sorting | 1.27–1.7 | Flattening | 1.9–8.7 |
| Total | 70.37–112 | Total | 8.5–38.9 |

producing ASR. The capital costs for buying the ASR separation and sorting equipment, and paying for other overheads, was estimated as 370,000 USD [22]. Furthermore, the energy, maintenance, labour, and interest costs can be up to 72,400 USD/yr. Assuming an income of at least 169,768 USD/yr, the investment can be returned in 3.8 years, where the average service life of the equipment is 12–15 years.

In the case of remanufacturing with a processing capacity of 232.5 t/d, this is equivalent to processing ~308 ELVs/d and remanufacturing 2,464 m$^2$/d of WSS. This would require the addition of three remanufacturing units to a dismantling facility which corresponds to the need for: 3 hydraulic cutting alligator machines (total of 25,107 USD), 3 plasma cutting machines (total of 62,768 USD), 3 expanding machines (total of 37,661 USD), and 3 flattening machines (total of 37,661 USD). The total investment for setting up the remanufacturing unit is around 163,200 USD, which is about half of that of an ASR plant. Assuming the income shown later in Table 8, this corresponds to a period of 180 d to return the investment, which is around 8 times shorter than that for ASR equipment.

## Remanufacturing feasibility assessment

This section discusses the feasibility of remanufacturing WSS into MSS based on the various calculated feasibility indexes described earlier. The technical, economic, and environmental feasibility are discussed separately, and then used to define an overall sustainability of the process.

**Technical feasibility.** Assuming $ER$ values of 1–14, $AR$ and the associated remanufacturing cost were calculated, as shown in Fig 3. An $ER$ of 8 gave the highest $AR$ of 11.5, while an $ER$ of 1 gave the lowest $AR$ of 2.5. Hence, the $ER$ and $AR$ can be optimized to produce remanufactured MSS for different purposes. As the processing cost is constant for every m$^2$ of WSS, the production cost per m$^2$ of produced MSS can be reduced by maximising $AR$. The lowest cost of 0.13 USD/m$^2$ was achieved for the highest $AR$ of 11.5. Considering the upper bound of required sales, 1 m$^2$ of remanufactured MSS can be sold for 0.2–0.6 USD/m$^2$, while the price of VSS is 4–5 USD/m$^2$, indicating that remanufacturing is economically feasible.

**Table 8. Calculated cost breakdown of the remanufacturing machinery operating costs.**

| Machine | No. machines | No. operators | Power cost (USD/d) | Space rent cost (USD/d) | Labour cost (USD/d) | Total cost (USD/d) |
|---|---|---|---|---|---|---|
| Hydraulic alligator cutter | 2 | 2 | 35 | 294 | 658 | 952 |
| Laser cutter | 3 | 8 | 47 | 294 | 2633 | 2974 |
| Hydraulic press brake | 4 | 2 | 35 | 294 | 658 | 987 |
| Mesh expander | 6 | 6 | 71 | 588 | 1975 | 2634 |
| Flattener | 2 | 2 | 35 | 294 | 658 | 987 |
| Leveller | 2 | 4 | 35 | 294 | 1316 | 1645 |
| Total | 19 | 24 | 223 | 2058 | 7898 | 10179 |

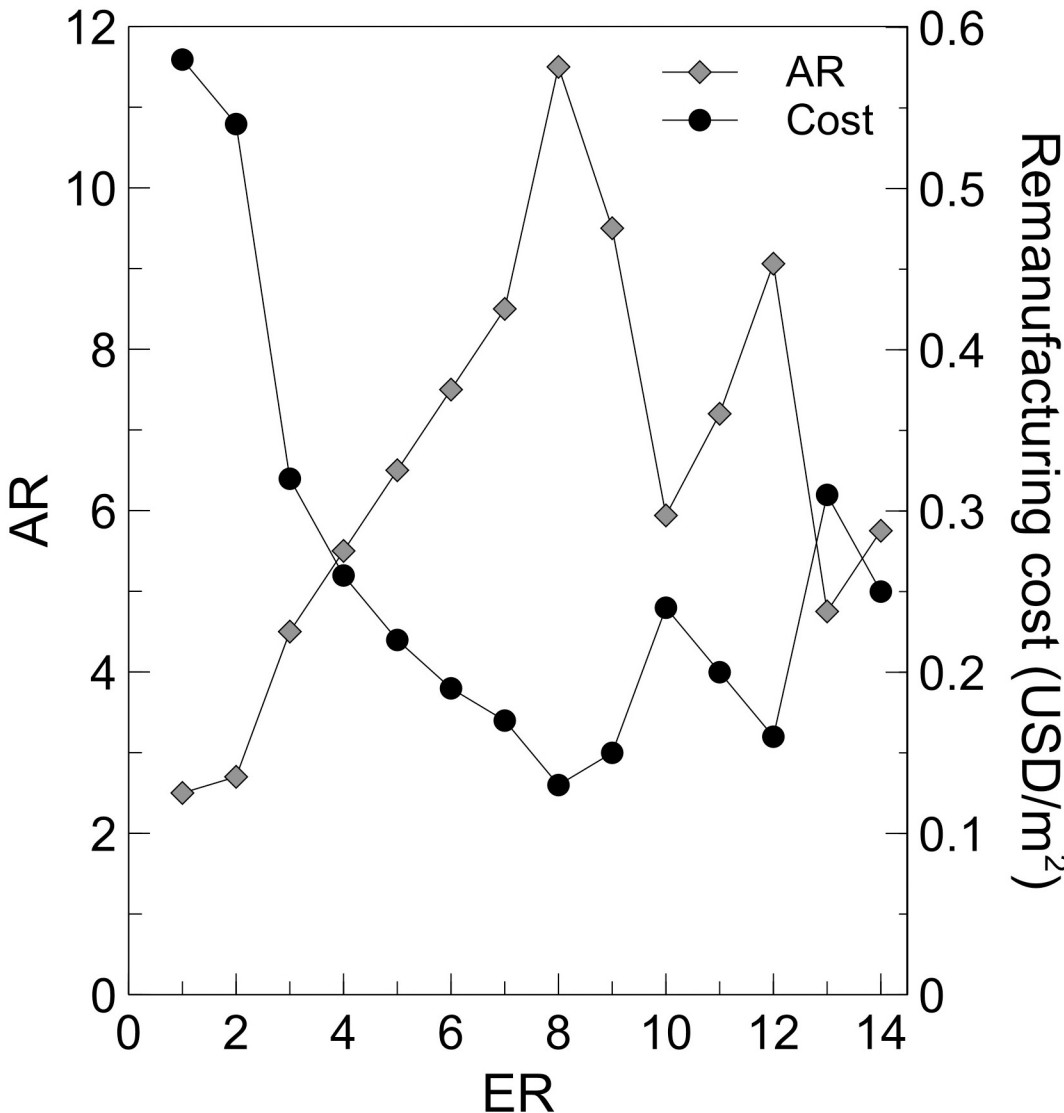

**Fig 3. Effect of the expansion ratio on the area gain ratio and remanufacturing cost.**

**Economic feasibility.** The productivity of a recycling plant can be increased by using dismantling lines, as proposed in previous studies [9, 10, 21], which could allow 27,600–31,171 vehicles to be processed per fabrication year (250 d). This would require the addition of 18 remanufacturing units to existing recycling plants with a productivity of 333–480 $m^2$/shift (assuming 24 h operation and 3 shifts per day, and 432 operators) [31]. The remanufacturing units equipped with the required machinery (as described previously) have much lower energy usage than the equipment required for smelting ASR. Table 8 shows the cost breakdown of the required machinery assuming a unit of productivity of 2851–5520 $m^2$/d of MSS.

Table 9 summarizes the sales of the same remanufacturing unit assuming MSS production of 2500–11,500 $m^2$/d and a sales price of 4.18 USD/$m^2$, resulting in sales of ~10,450–48,070 USD/d.

**Environmental feasibility.** The data used to calculate the environmental feasibility are shown in Table 10 (calculated using Eqs (12)–(14)). The ranges of values for the

**Table 9. Calculated remanufacturing sales.**

| Sales (USD/d) | | Cost (USD/d) | |
|---|---|---|---|
| DLB | 21426 | Rent | 4114 |
| DUB | 54755 | Labour | 15705 |
| IDLB | 42903 | Water | 518 |
| IDUB | 109642 | Total | 20427 |

remanufacturing process were calculated assuming the lower and upper bounds of productivity. The remanufacturing process resulted in reductions in $CO_2$ emissions and corresponding increase in $\varepsilon$ values of ~6–11 times higher than that of recycling, clearly indicating the huge environmental benefit of WSS remanufacturing. This analysis gave $E$ values of 0.832–0.913 (average of 0.873) for remanufacturing.

**Overall sustainability.** From Eqs (10), (11) and (14) the $T$, $C$, and $E$ values were calculated as 0.675, 0.834, and 0.873, respectively. To calculate the overall SI (Eq (15)), three scenarios were chosen with different weights for the technical, economic, and environmental feasibilities [9], as shown in Fig 4. In the first scenario, economic feasibility is more important than technical feasibility; in the second scenario, technical feasibility is more important than economic feasibility; and in the third scenario all three feasibilities are equally important. All three scenarios gave similar final SI values of 0.770–0.793 (average of 0.784). Previous studies [12–16] gave feasibility thresholds of $T \geq 0.7$, $C \geq 0.7$, and $E \geq 0.6$ for other remanufacturing processes. The $C$ and $E$ values determined here clearly exceeded the required thresholds, while $T$ was just under to threshold value. The overall $SI$ of 0.784 is considered sufficiently high to conclude that the proposed WSS–MSS remanufacturing process is a viable alternative to reuse and recycling procedures. There is a current move toward lightweight vehicle structures, with the aim of reducing fuel consumption. The total energy conservation can be further increased if coupled with adequate recycling management (to conserve about 51.0 MJ/vehicle), while incorrect combinations can increase energy consumption by 92.7 MJ/vehicle [32] and result in more material being sent to landfill. The introduction of WSS remanufacturing for recycling such vehicles could moderate this counterproductive effect, where power consumption could be reduced by ~230 MJ/vehicle, where the minimum power conservation threshold could be ~136 MJ/vehicle, even in the case where the recycling process is not ideal for lightweight ELV. In addition to the use of aluminium sheets for lightweight vehicles, which could also be suitable for remanufacturing into sheets, thin steel sheets (0.5–0.7 mm) are being used to fabricate exterior components to replace standard sheets with a thickness of 0.9–1 mm. Composite materials can be reused and remanufactured in the form of sheets without expanding. Further increases in the sustainability of vehicle recycling could be achieved by using high-capacity dismantling units with ability for vertical integration of vehicle manufacturers, end-of-life

**Table 10. Calculated environmental feasibility indicators.**

| Indicators | Recycling | Remanufacturing |
|---|---|---|
| Amount of VSS preserved (kg) | 3744 | 22,238–43,065 |
| Eco-cost/kg (USD) | 0.476 | 0.476 |
| $CO_2$ emission ($kg_{CO_2}/kg_{VSS}$) | 1.559 | 1.559 |
| $\varepsilon$ (USD/d) | 1782 | 10,585–20,495 |
| $CO_2$ reduction | 5837 | 34,669–67,138 |
| E | – | 0.832–0.913 |
| E average | – | 0.873 |

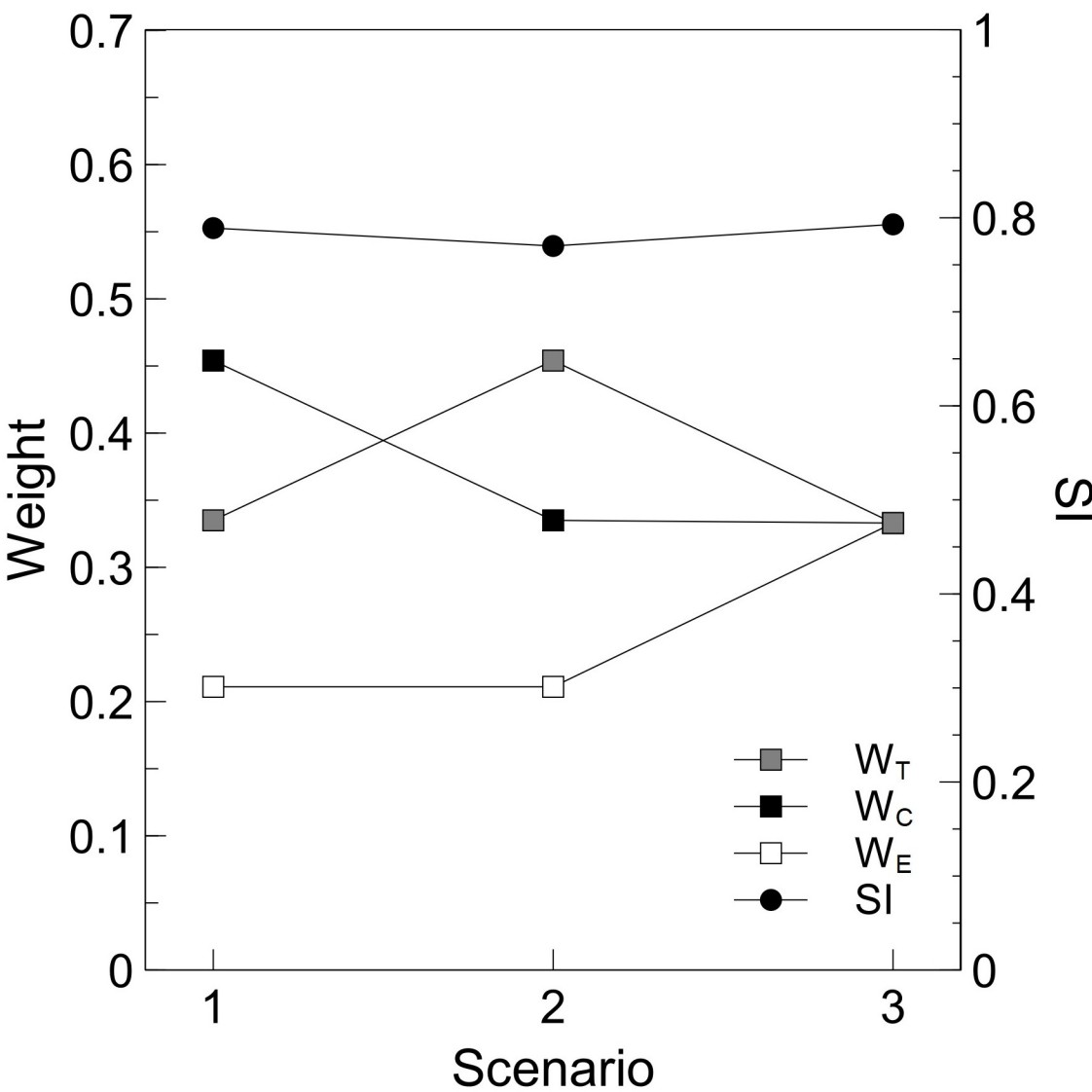

**Fig 4. Calculated sustainability indexes for three scenarios with different weights of importance for *T*, *C*, and *E*.**

management systems [32, 33], and horizontal integration of WSS remanufacturing units aligned with suitable policies.

## Conclusions

This study proposed an ELV remanufacturing process to improve both the eco-efficiency and profitability of the dismantling plant, where the production of MSS from WSS reduces power, cost, and carbon emissions compared to landfill disposal, combustion in municipal solid waste incinerators, and processing in advanced thermal treatment plants, while adding a profitable product to increase the economic viability of recycling. The remanufacturing process was concluded to be viable from the perspectives of the technical, economic, and environmental feasibility. Hence, it is recommended that such remanufacturing units could be integrated with existing disassembly plants to increase the overall sustainability and profitability of ELV recycling, where this model is particularly viable in developing countries as it is suitable for

small-to-medium enterprises with a modest investment and short payback period. In future studies, mixed integer programming models could be modified to include remanufacturing feasibility to provide data for vehicle production planning. In addition, the pricing problem could be evaluated using a nonlinear programming model to develop an approximate supply function for the ELVs with variable shredding costs to optimise the pricing considering the increasing costs of ferrous metals and ELVs.

## Author Contributions

**Conceptualization:** Ziyad Tariq Abdullah.

**Data curation:** Ziyad Tariq Abdullah.

**Formal analysis:** Ziyad Tariq Abdullah.

**Investigation:** Ziyad Tariq Abdullah.

**Methodology:** Ziyad Tariq Abdullah.

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
