## [Decision Letter · Decision Letter 0]

27 Sep 2021

PONE-D-21-27091Assessment of End-of-life Vehicle Recycling: Remanufacturing Waste Sheet Steel into Mesh SheetPLOS ONE

Dear Dr. Tariq,

Thank you for submitting your manuscript to PLOS ONE. After careful consideration, we feel that it has merit but does not fully meet PLOS ONE’s publication criteria as it currently stands. Therefore, we invite you to submit a revised version of the manuscript that addresses the points raised during the review process.

We look forward to receiving your revised manuscript.

Kind regards,

Huan Li

Academic Editor

PLOS ONE

Journal Requirements:

 [NO]. 

Reviewers' comments:

Reviewer's Responses to Questions

**Comments to the Author**

1. Is the manuscript technically sound, and do the data support the conclusions?

Reviewer #1: Yes

Reviewer #2: Partly

2. Has the statistical analysis been performed appropriately and rigorously? 

Reviewer #1: Yes

Reviewer #2: N/A

3. Have the authors made all data underlying the findings in their manuscript fully available?

Reviewer #1: Yes

Reviewer #2: Yes

4. Is the manuscript presented in an intelligible fashion and written in standard English?

Reviewer #1: Yes

Reviewer #2: Yes

5. Review Comments to the Author

Reviewer #1: 1. The manuscript assessed producing Mesh Sheet from recycled waste sheet steel.

2. The manuscript’s strength is based on data in the manufacturing plant in Iraq. The weakness is that it is unclear whether these findings can be applied to other regions since the assessment results are based on a specific manufacturing plant.

3. Provide a point-by-point list of my major recommendations for the improvement of the manuscript;

In Conclusion, include any quantitative findings from the evaluation.

The evaluation results should be graphed in addition to tables to facilitate visual understanding.

4. If necessary, provide a point-by-point list of my minor for the improvement of the manuscript.

On page 11, Line 4, the reference source is not found.

On page 12, Line 6, the reference source is not found.

On page 14, Line 3, the reference source is not found.

On page 15, Line 5, the reference source is not found.

On page 15, Line 44, what is the definition of ‘d’?

On page 17, Line 12, the reference source is not found.

On page 19, Line 22, the reference source is not found.

On page 20, Line 15, the reference source is not found.

On page 21, Line 6, the reference source is not found.

On page 23, Line 9, the reference source is not found.

On page 29, Line 11, MSS from MSS is not correct.

Reviewer #2: 1. Proper management of ELVs, including remanufacturing, recycling and reuse, is a necessary task towards sustainable development. It is worth making continuous efforts to enhance ELV management practices. The study presented in this manuscript is encouraged. It is hoped that the manuscript can be improved with the following suggestions.

2. Many missing links exist throughout the manuscript which are shown as “Error! Reference not found.” and must be corrected. These references, based on the context, may provide fundamental information for this study. Unable to refer to those references, the discussion developed in Method and Results sections seems to be interrupted.

3. (Epsilon) ecc, (Epsilon) rem and (Epsilon) rec are not introduced in the text. Thus, the rationale for Eq.(12) to (14) can not be understood.

4. Weights of engine and transmission consist of a large proportion of total weight of automobiles. It seems that the author counts their weights in estimating economical potential; however, only exterior components are considered in the feasibility calculation in the study. How are they considered in WSS? Is it practical to assume 80 m2 of sheet can be recovered by ~698 kg per car for remanufacturing as stated in the Results section?

5. Table 1 entitles “Weights of the components of WEPrD”; however, Table 1 contains weights for pre-disassembly, post-disassembly and expanding. The title of Table 1 may need to be revised. Besides, it is suggested to indicate the nature of values by notations, such as “estimated”, “calculated” and etc, if data presented in the table are generated in this study, to clarify whether those values are retrieved from literature or calculated results of this study.

6. Opinions from the author on differences among Reuse, recycling and remanufacturing of WSS may be explained more explicitly, regarding technical, economical and environmental feasibility. It is suggested to revise the discussion to illustrate the comparison among these processes from the viewpoint of the author.

7. There are some items in Results for discussion which were not mentioned in previous Introduction and Method sections, such as, remanufacturing feasibility assessment, and etc. It is suggested to explain their relationships with other indexes.

8. Quite a few paragraphs are used to discuss ASR and other materials in the disassembly plants. It is suggested to focus discussions on how WSS may contribute to MSS with technological or management strategies.

9. The manuscript is suggested to provide page numbers for both reviewing and reading purposes.

10. Most terms have full names followed by abbreviations when they first appear in the manuscript; however, there are still some missing. Also, some typos exist in the text. Please double check for accuracy.

6. PLOS authors have the option to publish the peer review history of their article (what does this mean?). If published, this will include your full peer review and any attached files.

Reviewer #1: **Yes: **Toshihiko Nakata

Reviewer #2: No

---

## [Author Response · Author response to Decision Letter 0]

16 Oct 2021

Rebuttal Letter for Manuscript PONE-D-21-27091" Assessment of End-of-life Vehicle Recycling: Remanufacturing Waste Sheet Steel into Mesh Sheet"

The responses to the reviewers’ comments appear here in red text. The changes in the revised manuscript also appear in red text.

Reviewer #1: 1. The manuscript assessed producing Mesh Sheet from recycled waste sheet steel.

2. The manuscript’s strength is based on data in the manufacturing plant in Iraq. The weakness is that it is unclear whether these findings can be applied to other regions since the assessment results are based on a specific manufacturing plant.

I apologize that this point was not clear in the original manuscript. The study is not based on a single physical plant, but on a model plant developed based on literature data worldwide. I have added a paragraph in the revised manuscript clarifying this point (page 5, lines 3-14).

3. Provide a point-by-point list of my major recommendations for the improvement of the manuscript;

In Conclusion, include any quantitative findings from the evaluation.

The evaluation results should be graphed in addition to tables to facilitate visual understanding.

The table format was chosen as there was a lot of data to include in the paper, and much of it is not appropriate for graphing (no obvious dependent and independent variables). In addition, the journal does not accept repetition of data in both graph and table form. However, if the reviewer can suggest some specific tables that they feel would be more informative in graph form, I will be happy to make this change. 

4. If necessary, provide a point-by-point list of my minor for the improvement of the manuscript.

On page 11, Line 4, the reference source is not found.

On page 12, Line 6, the reference source is not found.

On page 14, Line 3, the reference source is not found.

On page 15, Line 5, the reference source is not found.

On page 17, Line 12, the reference source is not found.

On page 19, Line 22, the reference source is not found.

On page 20, Line 15, the reference source is not found.

On page 21, Line 6, the reference source is not found.

On page 23, Line 9, the reference source is not found.

These link errors have been corrected.

On page 15, Line 44, what is the definition of ‘d’?

Here, the unit “/d” is per day. As “d” is the recommended SI unit/abbreviation for “day,” it has been used in this manuscript.

On page 29, Line 11, MSS from MSS is not correct.

Thank you for bringing this to my attention. I have made the change “MSS from WSS”.

Reviewer #2: 1. Proper management of ELVs, including remanufacturing, recycling and reuse, is a necessary task towards sustainable development. It is worth making continuous efforts to enhance ELV management practices. The study presented in this manuscript is encouraged. It is hoped that the manuscript can be improved with the following suggestions.

Thank you for the kind feedback. I hope that the following changes to the manuscript now make it suitable for publication.

2. Many missing links exist throughout the manuscript which are shown as “Error! Reference not found.” and must be corrected. These references, based on the context, may provide fundamental information for this study. Unable to refer to those references, the discussion developed in Method and Results sections seems to be interrupted.

These link errors have been corrected. These were references to the Figures and Tables. Apologies if this made it more difficult to review my manuscript.

3. (Epsilon) ecc, (Epsilon) rem and (Epsilon) rec are not introduced in the text. Thus, the rationale for Eq.(12) to (14) can not be understood.

Epsilon refers to the eco-cost saving. These symbols were indeed defined in the paragraph preceding Eq 12-14. Therefore, no change has been made in response to this comment.

4. Weights of engine and transmission consist of a large proportion of total weight of automobiles. It seems that the author counts their weights in estimating economical potential; however, only exterior components are considered in the feasibility calculation in the study. How are they considered in WSS? Is it practical to assume 80 m2 of sheet can be recovered by ~698 kg per car for remanufacturing as stated in the Results section?

The available WSS was calculated assuming that an average of 80 m2 can be recovered from each car to prevent manufacturing of new structural sheet steel of (~698 kg).

5. Table 1 entitles “Weights of the components of WEPrD”; however, Table 1 contains weights for pre-disassembly, post-disassembly and expanding. The title of Table 1 may need to be revised. 

I have revised the title of Table 1 to read “Calculated weights of the individual components used to calculate WEPrD, WEPoD, and WEEx” to clarify this.

Besides, it is suggested to indicate the nature of values by notations, such as “estimated”, “calculated” and etc, if data presented in the table are generated in this study, to clarify whether those values are retrieved from literature or calculated results of this study.

Thank you for this suggestion; this has been revised throughout the paper.

6. Opinions from the author on differences among Reuse, recycling and remanufacturing of WSS may be explained more explicitly, regarding technical, economical and environmental feasibility. It is suggested to revise the discussion to illustrate the comparison among these processes from the viewpoint of the author.

Some further discussion has been added on this point (page 13, line 16-22).

7. There are some items in Results for discussion which were not mentioned in previous Introduction and Method sections, such as, remanufacturing feasibility assessment, and etc. It is suggested to explain their relationships with other indexes.

The section titled “Remanufacturing feasibility assessment” discusses the results of the methods described in the section “Calculation of feasibility indexes”. I have added a sentence at the start of the former to explain this.

8. Quite a few paragraphs are used to discuss ASR and other materials in the disassembly plants. It is suggested to focus discussions on how WSS may contribute to MSS with technological or management strategies.

ASR is a current problem in the field, and its volume needs to be greatly reduced. Therefore, it is important to show the feasibility of remanufacturing WSS into MSS to increase the efficiency of dismantling, recycling, smelting, and landfill strategies by reducing ASR.

WSS is the raw material to remanufacture MSS, and since it is a solid waste management strategy, its sustainable development should include the dimensions of technical, economic, social, environmental, and management, which interact and interconnect to close the loop globally among the car industry and metal forming industry. This is important because many countries do not manufacture cars and cannot extend the Original Equipment Manufacturer responsibility to cope with end-of-life cars.

9. The manuscript is suggested to provide page numbers for both reviewing and reading purposes.

Page and line numbers have been added to aid reading and reviewing.

10. Most terms have full names followed by abbreviations when they first appear in the manuscript; however, there are still some missing. Also, some typos exist in the text. Please double check for accuracy. 

The entire paper has been revised again considering this advice.

---

## [Decision Letter · Decision Letter 1]

4 Nov 2021

PONE-D-21-27091R1Assessment of End-of-life Vehicle Recycling: Remanufacturing Waste Sheet Steel into Mesh SheetPLOS ONE

Dear Dr. Abdullah,

Thank you for submitting your manuscript to PLOS ONE. After careful consideration, we feel that it has merit but does not fully meet PLOS ONE’s publication criteria as it currently stands. Therefore, we invite you to submit a revised version of the manuscript that addresses the points raised during the review process.

We look forward to receiving your revised manuscript.

Kind regards,

Huan Li

Academic Editor

PLOS ONE

Journal Requirements:

Reviewers' comments:

Reviewer's Responses to Questions

**Comments to the Author**

1. If the authors have adequately addressed your comments raised in a previous round of review and you feel that this manuscript is now acceptable for publication, you may indicate that here to bypass the “Comments to the Author” section, enter your conflict of interest statement in the “Confidential to Editor” section, and submit your "Accept" recommendation.

Reviewer #1: (No Response)

Reviewer #2: (No Response)

2. Is the manuscript technically sound, and do the data support the conclusions?

Reviewer #1: Yes

Reviewer #2: Yes

3. Has the statistical analysis been performed appropriately and rigorously? 

Reviewer #1: Yes

Reviewer #2: N/A

4. Have the authors made all data underlying the findings in their manuscript fully available?

Reviewer #1: Yes

Reviewer #2: Yes

5. Is the manuscript presented in an intelligible fashion and written in standard English?

Reviewer #1: Yes

Reviewer #2: Yes

6. Review Comments to the Author

Reviewer #1: This decision was based on the following two points.

The table showing the results is merely a list of numbers, without any quantitative or qualitative analysis. There is a lot of value in organizing the results into graphs instead of tables, but the author does not do so.

The conclusion lacks the results of quantitative or qualitative analysis.

Reviewer #2: 1. Figure 1 and 2 are attached after the text should be provided with their Titles? According to the context, is Figure 1 supposed to show a remanufacturing process for converting WSS from ELVs into MSS?

2. In equation (14), (epsilon)�rec and �rem are functions of �ecc and �CO2, which are defined in equation(12) and (13) and they are explained in the paragraph. However, (epsilon)�rec and �rem must have different values of �ecc and �CO2 since their Mvss, Mco2, and �vss, right? My question is how are those values determined? They should be also be stated in this Methods section.

3. The estimation of 754 kg steel/ELV recycled from ~ 1 t/ELV, p10, includes weights of engine and transmission which are not considered as WSS. It should be distinguished clearly by using weights of vehicles with reuse, remanufacturing and recycling alternatives in discussion of the economical and environmental efficiencies in the manuscript.

7. PLOS authors have the option to publish the peer review history of their article (what does this mean?). If published, this will include your full peer review and any attached files.

Reviewer #1: No

Reviewer #2: No

---

## [Author Response · Author response to Decision Letter 1]

22 Nov 2021

Responses to the reviewer for Manuscript PONE-D-21-27091R1 "Assessment of End-of-life Vehicle Recycling: Remanufacturing Waste Sheet Steel into Mesh Sheet"

The responses to the reviewers’ comments appear here in red text. The changes in the revised manuscript also appear in red text.

Reviewer #1: This decision was based on the following two points.

The table showing the results is merely a list of numbers, without any quantitative or qualitative analysis. There is a lot of value in organizing the results into graphs instead of tables, but the author does not do so.

There are many tables listing the results, and it is a little unclear which one the reviewer is referring to. As many of the results have ranges of values, which are difficult to express in figure format, I chose the use of tables. In my previous response, I requested that the reviewer clarify which specific tables they would like to see in figure format to make this change. Without this information, it is difficult to comply with this request. As noted before, much of the data is not appropriate for graphing (no obvious dependent and independent variables), so graphing would make the data less easy to understand. 

I provide a summary of the tables and the reasons why most of them are better presented as tables. The only viable data for presentation in figure form (in my opinion) appears in Tables 8 and 12. Hence, I have plotted these data as figures in an attempt to address the reviewer’s concern.

Table 1: defines symbols and gives weight values used in calculations. These values are not results, and have no dependent variable.

Tables 2: it would be difficult to label the regions on a figure, and three different data sets are shown in one figure. It would be an inefficient use of space to plot these data. In addition, the calculated ranges would be difficult to express in a figure.

Table 3: as there is no common dependent variable, the comparison of the reuse and remanufacturing efficiencies is thought to be easier in table format.

Table 8: This table could be plotted as a figure. I have made this change (new Figure 3, as shown below).

Tables 4, 5, 6, 7, 9, 10, 11: No common dependent variable, many datasets in one table, inefficient to plot, ranges difficult to express in figures. 

Table 12: These data could be presented as a figure. I have made this change (new Figure 4, as shown below).

The conclusion lacks the results of quantitative or qualitative analysis.

I respectfully disagree with the reviewers comment, and apologize if the analysis was unclear. Without specific comments, it is difficult to address this concern. The main conclusion of the paper is related to the overall sustainability of remanufacturing waste sheet steel into mesh sheet steel. A detailed quantitative analysis was provided in the paper, which was condensed into an overall sustainability index as a final solution. This value is considered a quantitative conclusion. This simple way of comparing the various proposed remanufacturing scenarios provides a method for policy makers and ELV processing plants to evaluate the feasibility of the process. The paper provides a clear conclusion regarding the feasibility of the proposed process and its implications and potential benefits in the ELV processing field and automotive industry as a qualitative conclusion. 

Reviewer #2: 1. Figure 1 and 2 are attached after the text should be provided with their Titles? According to the context, is Figure 1 supposed to show a remanufacturing process for converting WSS from ELVs into MSS?

Following the formatting requirements of the journal, the figure captions were placed in the main text. You can find them on page 6 (lines 6-7) and page 7 (line 2).

2. In equation (14), (epsilon) rec and rem are functions of ecc and CO2, which are defined in equation(12) and (13) and they are explained in the paragraph. However, (epsilon) rec and rem must have different values of ecc and CO2 since their Mvss, Mco2, and vss, right? My question is how are those values determined? They should be also be stated in this Methods section.

Apologies if this was not clear in the original text. The terms εrem and εrec were calculated using Eq. (12) where εc is substituted by εrem or εrec as relevant. The reviewer is correct that MVSS is calculated using different values for each case, but εVSS and MCO2 are the same for both recycling and remanufacturing as they are based simply on the savings related to 1 kg of steel. The εc of VSS is 0.55 USD/kg and the MCO2 is 1.559 kgco2/kgVSS.

In the case of recycling 1 m2 of recovered WSS (assuming 100% recycling), the production of 7.8 kg of VSS, giving an MVSS value of 62.4 kg (assuming that the WSS is used for mesh-steel applications).

In the case of remanufacturing, 1 m2 of MSS can prevent 7.8 kg of VSS being produced, with upper and lower bounds of MVSS of 62.4 kg and 698 kg, respectively. The upper bound assumes that the steel is used for sheet-metal product applications and that the eco-design standards are satisfied.

Based on the number of end-of-life vehicles given in the scientific literature, the corresponding data for the recovered and remanufactured WSS was used to calculate the lower and upper bounds of economic feasibility in the form of eco-cost saving and CO2 prevention. 

I have added a discussion of this in the revised manuscript (page 11, lines 8-14).

3. The estimation of 754 kg steel/ELV recycled from ~ 1 t/ELV, p10, includes weights of engine and transmission which are not considered as WSS. It should be distinguished clearly by using weights of vehicles with reuse, remanufacturing and recycling alternatives in discussion of the economical and environmental efficiencies in the manuscript.

The value of 754 kg/ELV is the total amount of recyclable waste steel that can be recovered from an ELV (with a total weight of typically ~ 1 t). Further, 62.4 kg of the total 754 kg is the net weight of WSS which can be recovered, and its surface area can be amplified by 11.5 times. In the case of using MSS instead of VSS to produce sheet-metal-forming products, 1 m2 of MSS can save 1m2 of VSS which weighs 7.8 kg, so 698 kg of VSS is considered as the maximum resource conservation due to the possible area gain.

My apologies if this was not clear in the original text. The 745 kg quoted in the text does not include the weights of the engine and transmission, but only includes the weight of the virgin sheet steel which can be saved. Therefore, the analysis of the potential of the remanufacturing of WSS to MSS is only based on lightweight sheet steel. In the case of using MSS instead of VSS to produce sheet-metal-forming products, 1 m2 of MSS can save 1m2 of VSS which weighs 7.8 kg, so 745 kg of VSS is considered as the maximum resource conservation due to the possible area gain. I have carefully checked the phrasing throughout the paper to ensure that this is clear (page 10, line 5-8).

---

## [Editor Report · Decision Letter 2]

24 Nov 2021

Assessment of End-of-life Vehicle Recycling: Remanufacturing Waste Sheet Steel into Mesh Sheet

PONE-D-21-27091R2

Dear Dr. Abdullah,

We’re pleased to inform you that your manuscript has been judged scientifically suitable for publication and will be formally accepted for publication once it meets all outstanding technical requirements.

Kind regards,

Huan Li

Academic Editor

PLOS ONE
---

## [Editor Report · Acceptance letter]

29 Nov 2021

PONE-D-21-27091R2 

Assessment of End-of-life Vehicle Recycling: Remanufacturing Waste Sheet Steel into Mesh Sheet 

Dear Dr. Abdullah:

I'm pleased to inform you that your manuscript has been deemed suitable for publication in PLOS ONE. Congratulations! Your manuscript is now with our production department. 

Kind regards, 

on behalf of

Dr. Huan Li 

Academic Editor

PLOS ONE